Methods

# Laminin γ1 C-terminal Glu to Gln mutation induces early postimplantation lethality

Daiji Kiyozumi[1] , Yukimasa Taniguchi[1], Itsuko Nakano[1], Junko Toga[1], Emiko Yagi[1], Hidetoshi Hasuwa[2], Masahito Ikawa[2], Kiyotoshi Sekiguchi[1]

**Laminin–integrin interactions regulate various adhesion-dependent cellular processes. γ1C-Glu, the Glu residue in the laminin γ1 chain C-terminal tail, is crucial for the binding of γ1-laminins to several integrin isoforms. Here, we investigated the impact of γ1C Glu to Gln mutation on γ1-laminin binding to all possible integrin partners in vitro, and found that the mutation specifically ablated binding to α3, α6, and α7 integrins. To examine the physiological significance of γ1C-Glu, we generated a knock-in allele, Lamc1^EQ, in which the γ1C Glu to Gln mutation was introduced. Although Lamc1^EQ/EQ homozygotes developed into blastocysts and deposited laminins in their basement membranes, they died just after implantation because of disordered extraembryonic development. Given the impact of the Lamc1^EQ allele on embryonic development, we developed a knock-in mouse strain enabling on-demand introduction of the γ1C Glu to Gln mutation by the Cre-loxP system. The present study has revealed a crucial role of γ1C-Glu–mediated integrin binding in postimplantation development and provides useful animal models for investigating the physiological roles of laminin–integrin interactions in vivo.**

## Introduction

Laminins, αβγ trimeric glycoproteins, are major components of basement membranes (BMs) that play crucial roles in transmitting BM signals to cells. Various cellular processes including adhesion, migration, survival, proliferation, and differentiation are supported by laminins through interactions with cell-surface receptors. The most important receptors for sensing laminin signals are integrins, which are αβ dimeric cell-surface transmembrane proteins.

There are several modes of laminin–integrin interactions. One of the integrin-binding sites in laminins is located in a C-terminal αβγ complex known as the E8 fragment (Sonnenberg et al, 1990). The E8 fragments of laminins interact with multiple integrins, including α3β1, α6β1, α6β4, and α7β1 (Sonnenberg et al, 1990; Ido et al, 2007; Taniguchi et al, 2009). It is known that αβγ trimer formation, LG1–3 domains of the α chain, and Glu residue in the γ1 chain C-terminal tail (γ1C-Glu) (Fig 1A) are prerequisites for the integrin-binding ability of E8 fragments (Sung et al, 1993; Ido et al, 2004, 2007; Taniguchi et al, 2017). Recently, the crystal structures of truncated E8 fragments derived from laminin-111 and laminin-511 were solved (Pulido et al, 2017; Takizawa et al, 2017). The structures predicted that γ1C-Glu can bind directly to the metal ion-dependent adhesion site in the integrin β1 subunit. Thus, the biochemical significance of γ1C-Glu for integrin binding is apparent, but its physiological roles remain to be fully addressed.

Here, we investigated the physiological significance of γ1C-Glu for laminin–integrin interactions in vivo by generating a knock-in mouse strain in which γ1C-Glu was substituted with Gln. The resulting knock-in mice showed early postimplantation lethality, underscoring the critical role of γ1C-Glu–dependent laminin–integrin interactions for early embryonic development. Based on these findings, we established another knock-in mouse strain in which the γ1C Glu to Gln (EQ) mutation can be introduced on demand by the Cre-loxP system.

## Results and Discussion

### γ1 EQ mutation specifically abolishes laminin binding to α3, α6, and α7 integrins in vitro

Before starting in vivo studies, we confirmed the impact of γ1C-Glu on integrin binding by laminins in vitro (Fig 1). For this, we expressed and purified recombinant full-length laminin-111 and laminin-511 and their derivatives containing the EQ mutation (Fig S1). Because a panel of integrins including α1β1, α2β1, α3β1, α6β1, α6β4, α7β1, α9β1, αvβ3, and αvβ5 were reported to interact with these laminins (Forsberg et al, 1994; Pfaff et al, 1994; Sasaki & Timpl, 2001; Nishiuchi

[1]Laboratory of Extracellular Matrix Biochemistry, Institute for Protein Research, Osaka University, Osaka, Japan   [2]Research Institute for Microbial Diseases, Osaka University, Osaka, Japan

Correspondence: sekiguch@protein.osaka-u.ac.jp
Daiji Kiyozumi's present address is Immunology Frontier Research Center, Osaka University, Osaka, Japan
Hidetoshi Hasuwa's present address is Department of Molecular Biology, Keio University School of Medicine, Tokyo, Japan

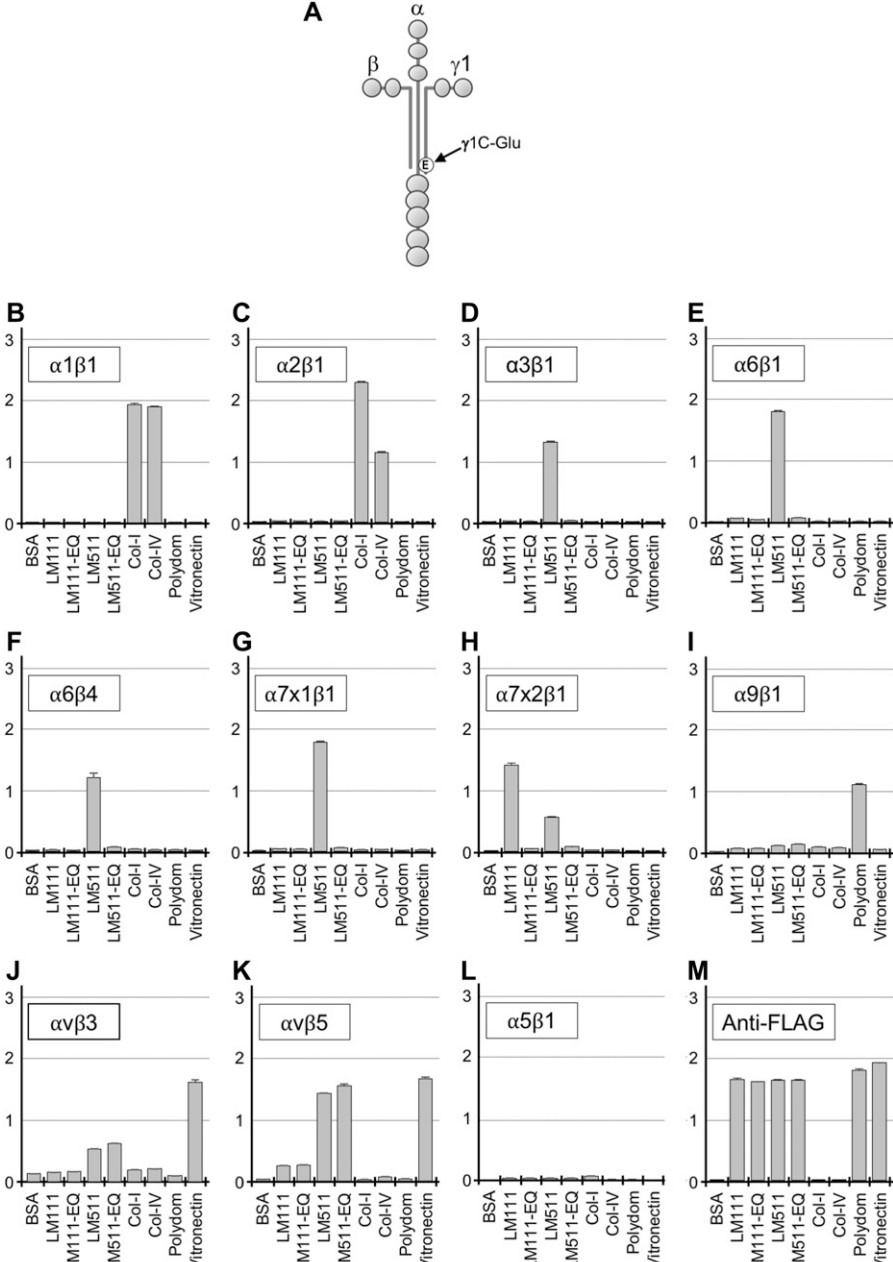

**Figure 1. Binding of recombinant laminin-111, laminin-511, and their EQ mutants to integrins.**
**(A)** Schematic diagram of a full-length laminin. The Glu (E) residue in the C-terminal region of the γ1 chain critical for integrin binding is indicated. **(B–L)** Binding of recombinant integrin α1β1 (B), α2β1 (C), α3β1 (D), α6β1 (E), α6β4 (F), α7x1β1 (G), α7x2β1 (H), α9β1 (I), αvβ3 (J), αvβ5 (K), and α5β1 (L) to immobilized laminins (LMs), type-I and type-IV collagens, polydom, and vitronectin. **(M)** Anti-FLAG antibody binding for quantification of immobilized recombinant proteins. Vertical axes represent absorbance at 490 nm which indicates integrin binding. Data represent means ± SD of triplicate assays.

et al, 2006), we expressed these integrins as truncated soluble forms in mammalian cells. Integrin α5β1 was also included as a negative control. The recombinant integrins were assessed for their abilities to bind to the laminins and their EQ mutants by solid-phase binding assays (Ido et al, 2007) (Figs 1B–M and S2A).

Integrins α3β1, α6β1, α6β4, and α7x1β1 bound specifically to laminin-511, whereas integrin α7x2β1 bound to both laminin-111 and laminin-511 (Fig 1D–H). Integrin α6β1 binding to recombinant laminin-111 was weak under the used condition (Fig 1E), unlike in previous reports that used Engelbreth-Holm-Swarm (EHS) tumor-derived mouse laminin-111 (Sonnenberg et al, 1990; Nishiuchi et al, 2003, 2006). When the coating concentration of laminin-111 was increased, integrin α6β1 was capable of binding to both mouse and

human laminin-111 in a dose-dependent manner, but mouse laminin-111 had significantly more affinity than human recombinant laminin-111 for integrin α6β1 (Fig S2B). Introduction of the γ1 EQ mutation abolished the abilities of laminin-111 and laminin-511 to bind to these integrins, consistent with our previous reports (Ido et al, 2007; Taniguchi et al, 2009). Among the other integrins examined, two Arg-Gly-Asp (RGD)-binding integrins, αvβ3 and αvβ5, exhibited significant binding to laminin-511 and weaker binding to laminin-111, and this binding was not compromised by the γ1 EQ mutation (Fig 1J and K). No binding to integrins α1β1, α2β1, α9β1, and α5β1 was observed (Figs 1B, C, I, L and S2A). This comprehensive survey of the integrin-binding activities of full-length laminin-111 and laminin-511 and their EQ mutants clearly showed that γ1

laminins were susceptible to the γ1 EQ mutation for their interactions with classical laminin-binding integrins, that is, α3β1, α6β1, α6β4, and α7β1.

## Generation of Lamc1[EQ] mice and embryonic lethality

To evaluate the impact of the γ1 EQ mutation in vivo, we generated a γ1 EQ knock-in allele, Lamc1[EQ], in which the codon encoding γ1C-Glu was mutated to introduce Gln (Fig 2A and B). Although Lamc1[EQ/+] heterozygous mice were fertile and did not exhibit any developmental defects, no Lamc1[EQ/EQ] neonates were obtained (Fig 2C).

Because αβγ trimer formation is a prerequisite for efficient secretion of laminins (Yurchenco et al, 1997), we examined whether laminins formed αβγ trimers in Lamc1[EQ/EQ] mice. Protein extracts

from Lamc1[EQ/EQ] E7.5 embryos were subjected to SDS–PAGE under nonreducing conditions and immunoblotted with an anti-laminin antibody. Lamc1[EQ/EQ] embryonic extracts produced a signal at ~800 kD similar to the case for Matrigel, a laminin-111–rich mouse tumor extract (Fig 2D). These results confirmed that EQ mutant laminins were able to form αβγ trimers. To address whether both laminin-111 and laminin-511 with a mutated γ1 chain can be secreted normally, we expressed these laminins in human 293 cells, and conditioned media were subjected to SDS–PAGE under non-reducing conditions and subsequent immunoblotting. No significant difference was detected in the amounts of the secreted heterotrimers between laminin-111/-511 and their EQ-mutants (Fig S3), suggesting that laminin-111 and laminin-511 with the mutated γ1 chain were secreted normally. Therefore, Lamc1[EQ/EQ] mice were phenotypically

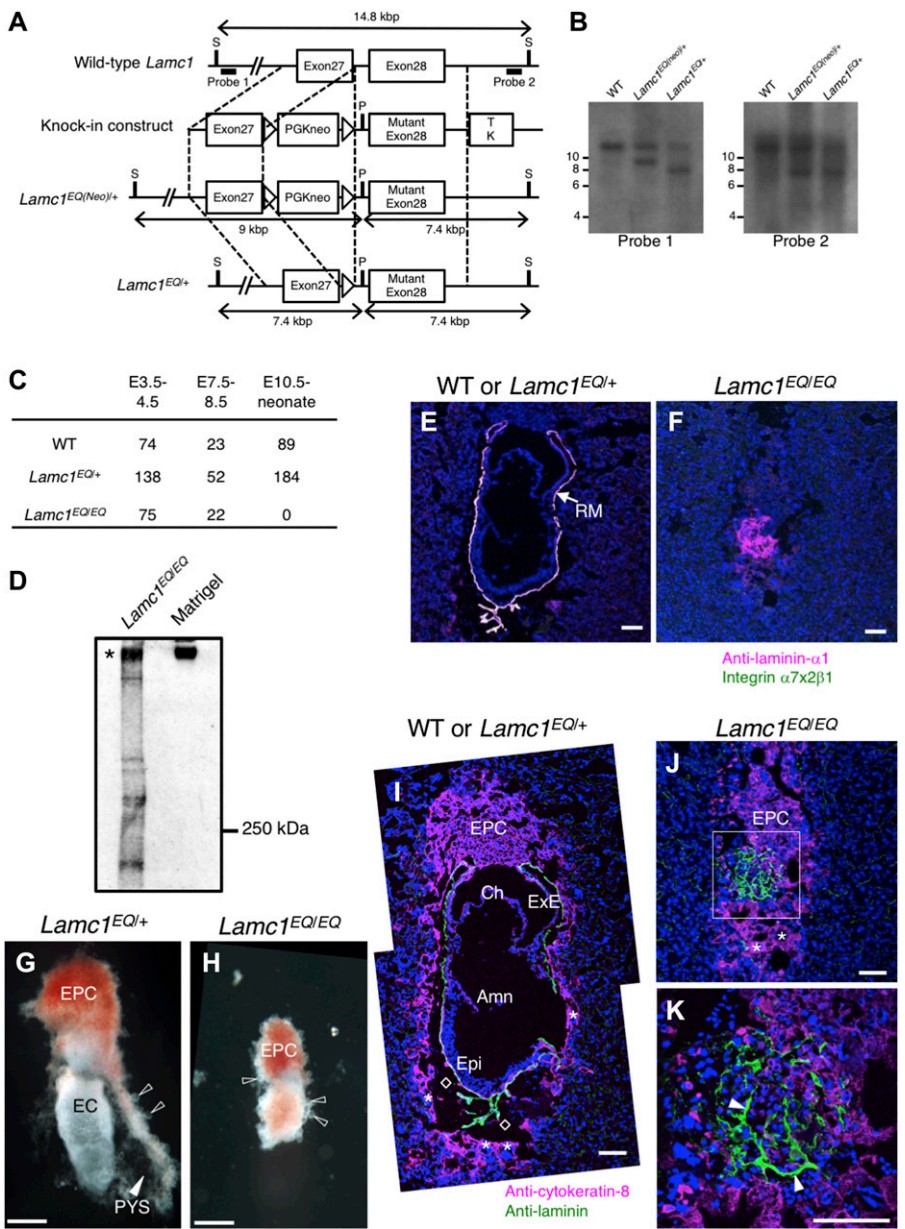

**Figure 2. Generation of Lamc1[EQ/EQ] knock-in mice.**
**(A)** Schematic diagram of the targeted mutation of Lamc1. The open boxes represent exons. The knock-in construct was designed to replace WT exon 28 encoding Glu at residue 1,605 with a mutant exon 28 encoding Gln at residue 1,605. The probes used for Southern blotting are indicated by bold lines. **(B)** Southern blot analyses of genomic DNA from WT, Lamc1[EQ(neo)/+], and Lamc1[EQ/+] offspring after digestion with SexAI and PacI. The detection of 9.0- and 7.4-kbp fragments with probe 1 and probe 2, respectively, in the Lamc1[EQ(neo)/+] lanes indicates the occurrence of the expected homologous recombination. The detection of a 7.4-kbp fragment with probe 1 in the Lamc1[EQ/+] lane indicates that the neomycin-resistance gene has been removed from the Lamc1[EQ(neo)] allele by the Cre-loxP system. **(C)** Survival of WT, Lamc1[EQ/+], and Lamc1[EQ/EQ] littermates obtained from Lamc1[EQ/+] intercrosses. **(D)** Western analyses of a protein extract from a Lamc1[EQ/EQ] E7.5 whole embryo and Matrigel using an anti-laminin antibody under nonreducing conditions. **(E, F)** In situ binding of recombinant integrin α7x2β1 (green) in E7.5 WT or Lamc1[EQ/+] (E) and Lamc1[EQ/EQ] (F) frozen sections. The RM was counterstained with an anti–laminin-α1 mAb (magenta). Bars, 100 μm. **(G, H)** Light microscopic images of control Lamc1[EQ/+] (G) and Lamc1[EQ/EQ] (H) E7.5 whole embryos. The filled arrowhead indicates the PYS. The open arrowheads indicate trophoblast giant cells. Bars, 200 μm. **(I–K)** Immunofluorescence staining for cytokeratin-8 (magenta) and laminin (green) in WT or Lamc1[EQ/+] (I) and Lamc1[EQ/EQ] (J, K) E7.5 sections. Blue, nuclei. The boxed area in J is magnified in K. The open diamonds in I indicate the blood sinus. The asterisks in I and J indicate trophoblast giant cells. The arrowheads in K indicate the extracellular matrix structure. Bars, 100 μm. Amn, amnion; Ch, chorion; ExE, extraembryonic ectoderm; Epi, epiblast; S, SexAI restriction site; P, PacI restriction site; TK, thymidine kinase gene; EPC, ectoplacental cone; EC, egg cylinder; PYS, parietal yolk sac.

different from $Lamc1^{-/-}$ mice (Smyth et al, 1999), because they expressed αβγ trimeric laminin, whereas $Lamc1^{-/-}$ mice did not. When E7.5 embryo sections were examined for binding of integrin α7x2β1, which can bind both laminin-111 and laminin-511 (Fig 1H), WT and $Lamc1^{EQ/+}$ embryos showed specific integrin α7x2β1 binding in their laminin-positive BMs (Fig 2E). $Lamc1^{EQ/EQ}$ homozygotes were distinguished from the other embryos by the absence of this integrin binding (Fig 2F). These results demonstrated that the γ1C-Glu–dependent integrin-binding activity of laminins was abolished by the γ1 EQ mutation in vivo.

To address the embryonic lethality of $Lamc1^{EQ/EQ}$ mice, E7.5 egg cylinders were investigated morphologically by light microscopy or sectioned and examined by immunofluorescence. Because laminin-111 and laminin-511 are expressed in Reichert's membrane (RM) (Sasaki et al, 2002; Miner et al, 2004), the parietal yolk sac, an extraembryonic tissue enveloping the egg cylinder, was visualized by its laminin-positive BM in sections of control WT/$Lamc1^{EQ/+}$ embryos (Fig 2G and I). However, a parietal sac was not observed in $Lamc1^{EQ/EQ}$ homozygotes, although laminin was expressed (Fig 2H, J, and K). Trophoblast giant cells, which are cytokeratin-8–positive and have large nuclei, were recognized in both WT/$Lamc1^{EQ/+}$ and $Lamc1^{EQ/EQ}$ embryos (Fig 2I and J), suggesting that extraembryonic cell differentiation was not affected in $Lamc1^{EQ/EQ}$ homozygotes. Collectively, $Lamc1^{EQ/EQ}$ mice showed morphogenetic defects at the early postimplantation stage.

## γ1C-Glu–dependent integrin binding is dispensable for BM deposition

To determine the timing for the morphological abnormality in $Lamc1^{EQ/EQ}$ homozygotes, we investigated the development of preimplantation embryos. Before implantation (E4.0 and E4.3), formation of the blastocoel cavity and inner cell mass were comparable between WT and $Lamc1^{EQ/EQ}$ blastocysts (Fig 3A–H). At E4.0, laminins were barely detected at the basal surface of the mural trophectoderm in blastocysts, irrespective of the genotypes (Fig 3A–D). Similarly, perlecan, another BM component, was not detected in E4.0 blastocysts (Fig 3A and B). In E4.3 blastocysts, laminins were detected at the basal surface of the mural trophectoderm in both WT and $Lamc1^{EQ/EQ}$ blastocysts (Fig 3E–H). Perlecan was also detected at the basal aspect of the mural trophectoderm in E4.3 blastocysts (Fig 3E and F). Primitive endoderm cells, the main laminin-producing cells producing dense signals with an anti-laminin antibody, were recognized in both WT and $Lamc1^{EQ/EQ}$ blastocysts (asterisks in Fig 3E–H). The laminin-binding integrin α6 was localized at the basal surface of the mural trophectoderm in WT and $Lamc1^{EQ/EQ}$ blastocysts irrespective of the laminin deposition in E4.0 (Fig 3C and D) and E4.3 (Fig 3G and H) blastocysts. To further examine the BM development in blastocysts, E3.5 blastocysts were flushed out from the uterus and cultured ex vivo for 48 h. In both ex vivo-cultured $Lamc1^{EQ/+}$ and $Lamc1^{EQ/EQ}$ blastocysts, laminins were densely deposited at the basal surface of the mural trophectoderm, where another BM component, perlecan, was colocalized (Fig 3I and J). These findings indicate that the preimplantation development of $Lamc1^{EQ/EQ}$ embryos was normal according to their morphology, laminin deposition, and BM formation. It has been reported that deficiency of BM formation in

embryoid bodies derived from β1 integrin-null mouse embryonic stem (ES) cells (Aumailley et al, 2000; Li et al, 2002) can be rescued by exogenous addition of laminin-111 in vitro (Li et al, 2002). Because the BM assembly rescued by exogenous laminin-111 was blocked by the addition of the E3 fragment of laminin-111, but not the E8 fragment, E3-binding non-integrin receptors (e.g., dystroglycan, syndecan, and sulfated glycolipids) play dominant roles in the BM assembly of laminins. The dispensability of integrin binding of laminins in BM formation shown here using ex vivo-cultured $Lamc1^{EQ/EQ}$ blastocysts is consistent with these previous findings. Nevertheless, we cannot exclude the possibility that γ1 EQ mutation affects the BM assembly of laminins, which is not discernible at the level of conventional histology, thereby contributing to the phenotype of $Lamc1^{EQ/EQ}$ embryos.

Because no apparent abnormalities in morphology and laminin deposition were observed in the $Lamc1^{EQ/EQ}$ blastocysts, we investigated the early postimplantation development at E5.5 by immunofluorescence (Fig 3K–O). Expression of Oct4 and Cdx2, markers for the epiblast and extraembryonic ectoderm, respectively, was detected in $Lamc1^{EQ/EQ}$ embryos and in the control WT/$Lamc1^{EQ/+}$ littermates (Fig 3M and N). Visceral endoderm and parietal endoderm cells were also observed in $Lamc1^{EQ/EQ}$ sections (Fig 3O). Thus, both embryonic and extraembryonic cell fate decisions occurred in $Lamc1^{EQ/EQ}$ embryos. RM, which was stained with an anti-laminin-α1 antibody, was formed in control WT and $Lamc1^{EQ/+}$ heterozygotes at E5.5 (Fig 3K). RM was also formed in $Lamc1^{EQ/EQ}$ homozygotes at E5.5 (Fig 3L), but showed severe disorganization by E7.5 (Fig 2J and K). These findings indicate that RM formation was not affected in $Lamc1^{EQ/EQ}$ mice until at least E5.5. It was previously shown in vivo that RM formation requires α1-containing laminins and dystroglycan (Williamson et al, 1997; Miner et al, 2004). However, the deficient RM formation observed in $Lamc1^{EQ/EQ}$ mice was different from that in laminin-α1 or dystroglycan knockout mice because the RM was initially formed in $Lamc1^{EQ/EQ}$ mice.

## γ1C-Glu–dependent integrin binding is indispensable for postimplantation development of extraembryonic tissues

Although cell differentiation and BM formation appeared unaffected at E5.5, $Lamc1^{EQ/EQ}$ embryos were smaller than control WT/$Lamc1^{EQ/+}$ embryos (Fig 3M and N) and exhibited poor development of extraembryonic tissues. The parietal yolk sac in $Lamc1^{EQ/EQ}$ embryos also appeared smaller than that in WT/$Lamc1^{EQ/+}$ embryos (Fig 3M and N). Therefore, we investigated the development of the parietal yolk sac by measuring the RM length on sections by quantitative image analysis (Fig 3P and Q). After implantation at E5.5, the parietal yolk sac in WT and $Lamc1^{EQ/+}$ zygotes grew expansively and the RM length reached 452 ± 83 μm when measured on sections. By contrast, the RM length in $Lamc1^{EQ/EQ}$ homozygotes was 217 ± 41 μm at E5.5, being significantly smaller than that in control embryos. These findings clearly showed that the expansive growth of the parietal yolk sac was defective in $Lamc1^{EQ/EQ}$ homozygotes. Interestingly, the extraembryonic ectoderm was dissociated from its BM in $Lamc1^{EQ/EQ}$ E5.5 zygotes (Fig 3N), whereas that in the control WT/$Lamc1^{EQ/+}$ zygotes was tightly associated with the BM (Fig 3M). The embryonic part of $Lamc1^{EQ/EQ}$ embryos

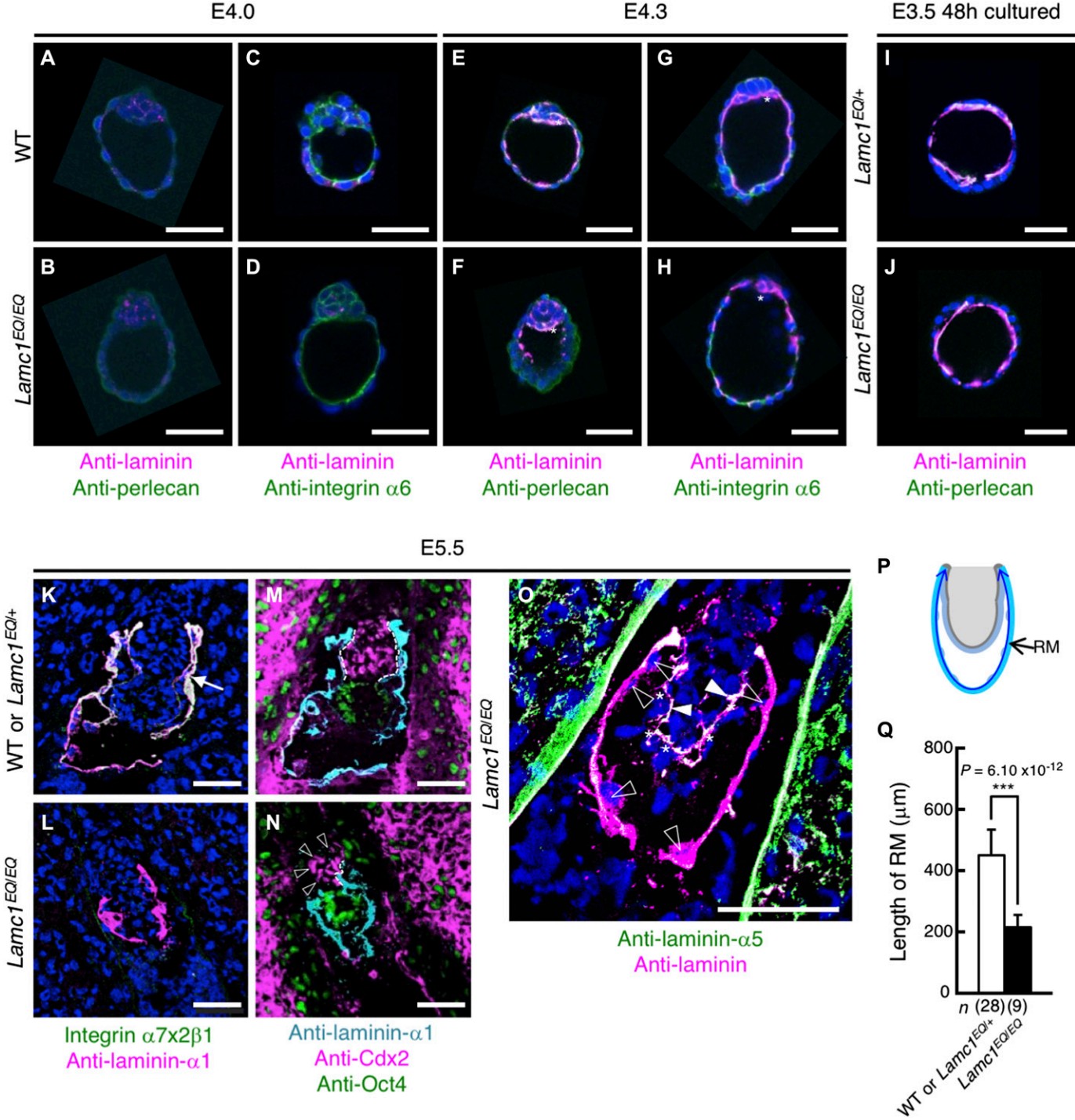

**Figure 3. BM formation in _Lamc1_<sup>EQ/EQ</sup> zygotes.**
**(A–H)** Whole-mount immunostaining of WT (A, C, E, G) and _Lamc1_<sup>EQ/EQ</sup> (B, D, F, H) blastocysts of E4.0 (A–D) and E4.3 (E–H) embryos. The asterisks indicate primitive endoderm cells. **(I, J)** Whole-mount immunostaining of E3.5 _Lamc1_<sup>EQ/+</sup> (I) and _Lamc1_<sup>EQ/EQ</sup> (J) blastocysts cultured ex vivo for 48 h. Magenta, laminin; green, perlecan (A, B, E, F, I, J) or integrin α6 (C, D, G, H); blue, nuclei. Bars, 50 μm. The genotypes of the preimplantation embryos were determined by genomic PCR. **(K, L)** In situ binding of integrin α7x2β1 (green) to E5.5 embryonic sections. The RM was counterstained with an anti-laminin-α1 mAb (magenta). Blue, nuclei. **(M, N)** Immunofluorescence staining for laminin-α1 (cyan), Cdx2 (magenta), and Oct3 (green) in WT or _Lamc1_<sup>EQ/+</sup> (M) and _Lamc1_<sup>EQ/EQ</sup> (N) E5.5 sections. The dashed lines indicate the region where extraembryonic ectoderm cells are associated with the BM. The open arrowheads indicate extraembryonic ectoderm cells dissociated from the BM. **(O)** Immunofluorescence staining for laminin (cyan) and laminin-α5 (green) in _Lamc1_<sup>EQ/EQ</sup> E5.5 sections. Blue, nuclei. The asterisks and open triangles indicate visceral endoderm cells and parietal endoderm cells, respectively. The arrowheads indicate the laminin-α5–positive epiblast BM. Bars, 50 μm. **(P)** Summary of the strategy for quantitative image analysis of RM length in the E5.5 egg cylinder. The RM length (double-headed arrow) was measured by image tracing. **(Q)** RM lengths measured in E5.5 _Lamc1_<sup>EQ/EQ</sup> embryos and control WT or _Lamc1_<sup>EQ/+</sup> embryos. Data represent means ± SD. ***P < 0.001, significant difference by Welch's _t_ test. The effect size was 3.13 and the statistical power was 1.0. ExE, extraembryonic ectoderm; Epi, epiblast.

was not critically affected, as the epiblast and visceral endoderm remained associated with their laminin-α5–positive BMs (Fig 3O). These results indicated that postimplantation development of extraembryonic tissues was critically dependent on laminin–integrin interactions through γ1C-Glu. No single α3, α6, or α7 integrin knockout mice were reported to exhibit early post-implantation defects (Georges-Labouesse et al, 1996; Kreidberg et al, 1996; Flintoff-Dye et al, 2005). Because integrin α3 and α6 are dispensable for early postimplantation development, as shown by α3, α6 double-knockout mice (De Arcangelis et al, 1999) early postimplantation development is probably secured by cooperative functioning of α7 integrin with α3 and/or α6 integrins. It is, therefore, likely that integrin α7 functions cooperatively with α3 and/or α6 integrins in extraembryonic tissues because integrin α7 is expressed in the trophectoderm-derived extraembryonic tissues (Klaffky et al, 2001).

### Conditional γ1 EQ knock-in mice as a tool for investigating the roles of laminin–integrin interactions in adult tissues

The above results clearly showed that the $Lamc1^{EQ}$ knock-in allele is a valuable tool for investigating the roles of laminin–integrin interactions in vivo, although homozygotes die at the early post-implantation stage. Based on these results, we tried to develop a conditional knock-in mouse strain in which γ1C-Glu–dependent integrin binding can be ablated on demand. By homologous recombination in ES cells, we generated another $Lamc1$ mutant allele, $Lamc1^{cEQ}$, in which exon 28 was floxed and an additional EQ-mutated exon 28 was located in its 3′ downstream (Fig 4A and B). $Lamc1^{cEQ/cEQ}$ homozygous mice were obtained according to the expected Mendelian ratio (Fig 4C) and appeared healthy and fertile.

Because the γ1 EQ knock-in mice showed embryonic lethality at the early developmental stage (Fig 2C), we determined whether tissue-specific ablation of γ1C-Glu–dependent laminin–integrin interactions is available while preventing embryonic lethality. We introduced the $Tie2$-$cre$ transgene (Kisanuki et al, 2001), expressing Cre recombinase in an endothelial cell-specific manner, into the $Lamc1^{cEQ/cEQ}$ genetic background, and examined whether integrin binding to laminins was ablated in blood vessels. When adult retinal sections were probed with recombinant integrin α3β1, recombinant integrin binding to blood vessel BMs was observed in control $Lamc1^{cEQ/cEQ}$ mice (Fig 4D–F), but not in $Tie2$-$cre$;$Lamc1^{cEQ/cEQ}$ mice (Fig 4G–I), consistent with blood vessel–specific introduction of the γ1 EQ mutation. As $Tie2$-$cre$;$Lamc1^{cEQ/cEQ}$ mice were obtained according to the expected Mendelian ratio (Fig 4J), conditional ablation of γ1C-Glu–dependent laminin–integrin binding in vivo was shown to be feasible.

Here, we have revealed the significance of γ1C-Glu in the interactions of laminins with α3, α6, and α7 integrins by in vitro binding assays and by generating γ1 EQ knock-in mice. We have further developed a conditional γ1 EQ knock-in system in mice. Although laminin–integrin interactions have been extensively studied by in vitro analyses at the molecular and/or cellular levels, it has been difficult to verify the in vitro results in vivo. Our conditional γ1 EQ knock-in system in mice will be a valuable tool for investigating the roles of laminin–integrin interactions in various physiological and pathological situations in vivo.

## Materials and Methods

### Antibodies and reagents

Rat monoclonal antibodies (mAbs) against mouse laminin α1 (5B7-H1) and α5 (M5N8-C8) were produced as described (Manabe et al, 2008). Mouse mAbs against human laminin α1 (5A3), α5 (15H5), β1 (DG10), and γ1 (C12SW) were produced as described previously (Kikkawa et al, 1998; Ido et al, 2007). Mouse mAbs against human laminin β1 were purchased from Enzo Life Sciences. Rabbit poly-clonal antibody (pAb) against Velcro (ACID/BASE coiled-coil) peptides was produced as described (Takagi et al, 2001). Mouse anti-FLAG mAb, rabbit anti-laminin pAb, and BSA were obtained from Sigma-Aldrich; rat anti-perlecan mAb was purchased from Merck Millipore; rat anti-mouse integrin-α6 mAb (GoH3) was from BD Biosciences; mouse anti-Cdx2 mAb was from Biocare; rabbit anti-Oct4 pAb was from Santa Cruz Biotechnology; HRP-conjugated donkey anti-mouse IgG and Cy3-conjugated anti-rat IgG were from Jackson ImmunoResearch; Alexa 488-conjugated goat anti-rabbit IgG, Alexa 546-conjugated goat anti-rat IgG, Alexa 405-conjugated goat anti-rabbit IgG, Alexa 488-conjugated goat anti-rat IgG, and Alexa 546-conjugated goat anti-mouse IgG were from Invitrogen; HRP-conjugated streptavidin was from Thermo Fisher Scientific. Rat anti-cytokeratin-8 mAb (TROMA-I), developed by Philippe Brulet and Rolf Kemler, was obtained from the Developmental Studies Hybridoma Bank. The antibody dilutions used are shown in Table S1. Bovine type I and type IV collagens were obtained from Nippi Inc. Mouse laminin-111 was prepared from mouse Engelbreth-Holm-Swarm tumor as described previously (Nishiuchi et al, 2006). Fi-bronectin was purified from human plasma by gelatin affinity chromatography as described previously (Sekiguchi & Hakomori, 1983).

### Expression vectors

Expression vectors for human laminin α1, α5, β1, and FLAG-tagged γ1 chains were prepared as described (Hayashi et al, 2002; Ido et al, 2004, 2006, 2008). The FLAG tag of laminin γ1 was located just after the N-terminal signal peptide cleavage site to facilitate purification of recombinant laminins. Expression vectors for the extracellular domains of the human integrin α1, α2, α3, α6, α7x1, α7x2, and α9 subunits were constructed as described (Nishiuchi et al, 2006; Sato-Nishiuchi et al, 2012; Jeong et al, 2013). Expression vectors for the extracellular domains of the human integrin αv, β1, β3, and β4 subunits were kindly provided by Dr. Junichi Takagi (Institute for Protein Research, Osaka University) (Takagi et al, 2001, 2002a, b). An expression vector for the extracellular domains of human integrin α5 was constructed in a similar manner to those of other integrin α subunits (Nishiuchi et al, 2006; Sato-Nishiuchi et al, 2012; Jeong et al, 2013). An expression vector for the extracellular domain of human integrin β5 was prepared using a cDNA amplified from RNA extracted from T98G human glioblastoma cells. Expression vectors

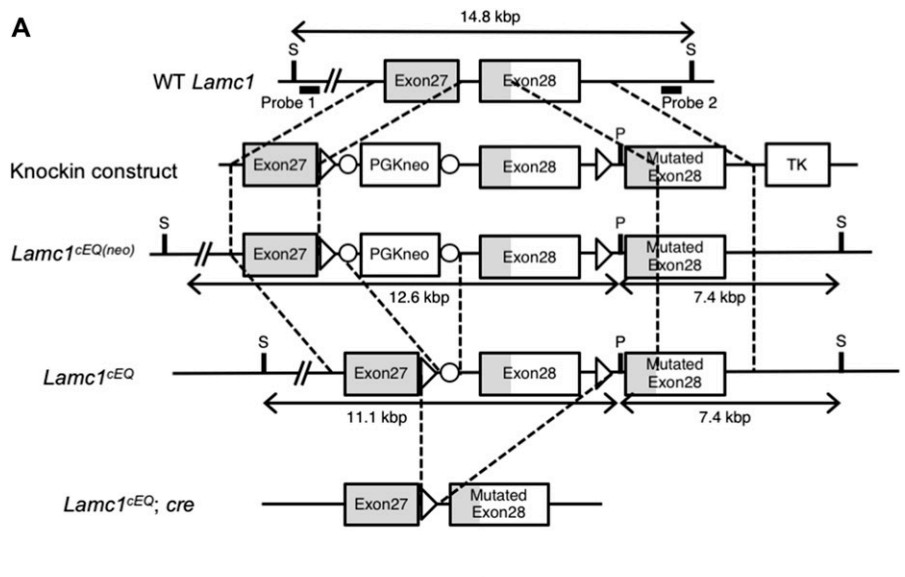

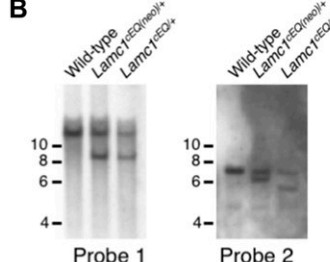

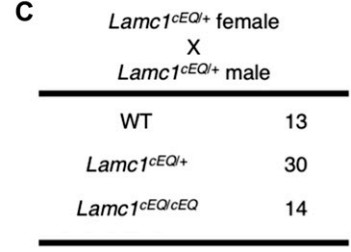

**Figure 4. Generation of γ1 EQ conditional knock-in mice.**
**(A)** Schematic representation of the generation of *Lamc1cEQ*. The open boxes represent exons. The protein coding sequences are indicated in gray. The targeting construct was designed to replace WT exon 28 encoding Glu at residue 1,605 with a floxed exon 28 followed by a mutated exon 28 encoding Gln at residue 1,605. The probes used for Southern blotting are indicated by bold lines. **(B)** Southern blot analyses of genomic DNA from WT, *Lamc1cEQ(neo)/+*, and *Lamc1cEQ/+* offspring after digestion with SexAI and PacI. The detection of 9.0 and 7.4 kbp fragments with probe 1 and probe 2, respectively, in the *Lamc1cEQ(neo)/+* lanes indicates occurrence of the expected homologous recombination. The detection of a 7.4 kbp fragment with probe 1 in the *Lamc1cEQ/+* lane indicates that the neomycin-resistance gene has been removed from the *Lamc1cEQ(neo)* allele by the Cre-loxP system. **(C)** Genotypes of offspring obtained from *Lamc1cEQ/+* intercrosses. **(D–I)** Histochemical analyses of *Lamc1cEQ/cEQ* (D–F) and *Tie2-cre;Lamc1cEQ/cEQ* (G–I) retinas. **(D, G)** In situ binding of recombinant integrin α3β1 (magenta) to frozen retinal sections. **(E, H)** Counterstaining of vascular BM with an anti–laminin-α5 antibody (green). Merged images with nuclear staining (blue) are also shown (F and I). Retinal vasculatures are indicated by filled (*Lamc1cEQ/cEQ*) and open (*Tie2-cre;Lamc1cEQ/cEQ*) arrowheads. Bars, 50 μm. **(J)** Genotypes of offspring obtained from mating between *Lamc1cEQ/+* female and *Tie2-cre;Lamc1cEQ/+* male mice. Only *Lamc1cEQ/cEQ* and *Tie2-cre;Lamc1cEQ/cEQ* mice are shown. S, SexAI restriction site; P, PacI restriction site; TK, thymidine kinase gene.

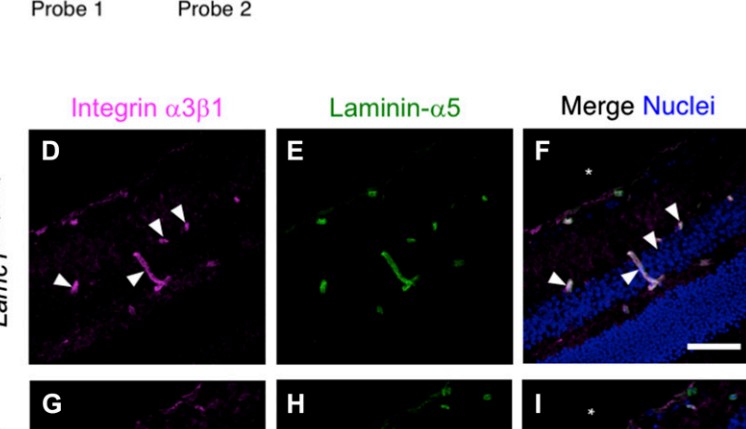

Integrin α3β1 Laminin-α5 Merge Nuclei

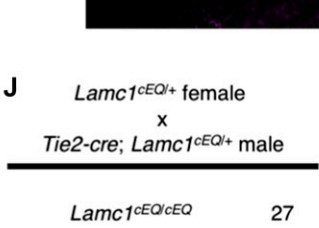

for recombinant human vitronectin with an N-terminal FLAG tag and the C-terminal fragment of mouse polydom were prepared as described (Sato-Nishiuchi et al, 2012; Ozawa et al, 2016).

### Expression and purification of recombinant proteins

Recombinant human laminin-111, laminin-511, and their EQ mutants were produced using a FreeStyle 293 Expression System (Thermo Fisher Scientific) as described (Ido et al, 2004). The conditioned media were passed over ANTI-FLAG M2 Affinity Gel (Sigma-Aldrich). After washing with 20 mM Tris-buffered saline without divalent cations (TBS), bound proteins were eluted with 100 $\mu$g/ml FLAG peptide (Sigma-Aldrich) and dialyzed against TBS. Purified proteins were verified by 4% SDS–PAGE under reducing conditions (Fig S1). Other recombinant proteins (integrin $\alpha1\beta1$, $\alpha2\beta1$, $\alpha3\beta1$, $\alpha6\beta1$, $\alpha7x1\beta1$, $\alpha7x2\beta1$, $\alpha9\beta1$, $\alpha v\beta3$, $\alpha v\beta5$, and $\alpha5\beta1$, and vitronectin and C-terminal fragment of polydom) were produced using the FreeStyle 293 Expression System and purified on ANTI-FLAG M2 Affinity Gel as described above. Protein concentrations were determined with a BCA protein assay kit (Thermo Fisher Scientific) using BSA as a standard.

### Laminin secretion assay

Recombinant human laminin-111, laminin-511, and their EQ mutants were produced using the FreeStyle 293 Expression System as described above. Conditioned media were recovered at 3 d after transfection and applied to SDS–PAGE and subsequent Western blot analyses.

### Western blotting

E7.5 embryos were lysed in 20 $\mu$l of SDS–PAGE sample buffer under nonreducing conditions. Purified laminins were prepared as samples for SDS–PAGE under reducing conditions. Conditioned media of human 293 cells were mixed with equal amounts of 2× SDS–PAGE sample buffer under nonreducing conditions. Proteins were separated by 4% SDS–PAGE and transferred onto nitrocellulose membranes. The membranes were immunoblotted with the indicated primary Abs and goat anti-rabbit or anti-mouse IgG secondary antibodies conjugated with HRP (Jackson Immuno-Research Laboratories). The bound primary antibody was visualized with an Amersham ECL Prime Western blotting detection reagent kit (GE Healthcare).

### Integrin binding assays

Solid-phase assays for binding of integrins to laminins, collagens, vitronectin, and polydom were carried out as described (Taniguchi et al, 2017). Briefly, 96-well microtiter plates were coated with recombinant proteins (laminins, vitronectin, and polydom, 5 nM; collagens, 10 $\mu$g/ml) overnight at 4°C, blocked with 3% BSA for 1 h at room temperature, and incubated with 30 nM integrins for 3 h at room temperature in the presence of 1 mM $Mn^{2+}$. Bound integrins were detected after sequential incubations with biotinylated rabbit anti-Velcro pAb and HRP-conjugated streptavidin. The amounts of recombinant laminins, vitronectin, and polydom adsorbed on the plates were quantified by enzyme-linked immunosorbent assays using an anti-FLAG M2 mAb to confirm equality of the adsorbed proteins.

### Mice

The B6.Cg-Tg(CAG-cre)CZ-MO2Osb mouse strain expressing Cre-recombinase under the CAG promoter (BRC No. 01828) was provided by the RIKEN BioResource Center with support from the National BioResource Project of the Ministry of Education, Culture, Sports, Science and Technology, Japan. The B6-Tg(CAG-FLPe)36 mouse strain expressing Flp-recombinase under the CAG promoter (BRC No. 01834) (Kanki et al, 2006) was also provided by RIKEN BioResource Center with support from the National BioResource Project of the Ministry of Education, Culture, Sports, Science and Technology, Japan. The B6.Cg-Tg(Tek-cre)1Ywa/J (*Tie2-cre*) mouse strain expressing Cre-recombinase under the Tek (Tie2) promoter (Kisanuki et al, 2001) was provided by Jackson Laboratory.

To generate *Lamc1$^{EQ}$* knock-in mice, a knock-in vector was constructed to include the 2-kb upstream *Lamc1* genomic sequence, a neomycin-resistance gene sandwiched by loxP sequences, the 5-kb downstream *Lamc1* genomic sequence, and a thymidine kinase gene, as shown schematically in Fig 2A. To introduce the γ1 EQ mutation, the 5-kb genomic sequence was point-mutated by overlap-extension PCR. The knock-in vector was introduced into strain 129 mouse ES cells. The resulting targeted clones were verified by PCR and Southern blotting and injected into C57Bl/6 blastocysts to obtain chimeric mice. Male chimeric mice that transmitted the mutated *Lamc1* gene through the germline were crossed with C57Bl/6 female mice (Japan SLC) to generate *Lamc1$^{EQ(Neo)}$* mice. The *Lamc1$^{EQ(Neo)/+}$* heterozygotes were crossed with B6.Cg-Tg(CAG-cre)CZ-MO2Osb mice to obtain *Lamc1$^{EQ/+}$* heterozygotes, in which the neomycin-resistance gene was removed. The *Lamc1$^{EQ/+}$* heterozygotes were crossed to obtain *Lamc1$^{EQ/EQ}$* homozygotes and WT littermates. E0.5 was defined as noon on the day of plug detection.

To generate *Lamc1$^{cEQ}$* mice, a knock-in vector was constructed as follows. The vector included the 2-kb upstream *Lamc1* genomic sequence, a neomycin-resistance gene sandwiched by loxP sequences, the 5-kb downstream *Lamc1* genomic sequence, and a thymidine kinase gene, as shown schematically in Fig 4A. To introduce the γ1 EQ mutation, the 5-kb genomic sequence was point-mutated by overlap-extension PCR. The knock-in vector was introduced into strain 129 mouse ES cells. The resulting targeted clones were verified by PCR and Southern blotting, and injected into C57Bl/6 blastocysts to obtain chimeric mice. Male chimeric mice that transmitted the mutated *Lamc1* gene through the germline were crossed with C57Bl/6 female mice to generate *Lamc1$^{cEQ(Neo)}$* mice. The resulting *Lamc1$^{cEQ(Neo)/+}$* heterozygotes were crossed with B6-Tg(CAG-FLPe)36 mice to obtain *Lamc1$^{cEQ/+}$* heterozygotes in which the neomycin-resistance gene was removed. *Lamc1$^{cEQ/+}$* heterozygotes were crossed to obtain *Lamc1$^{cEQ/cEQ}$* homozygotes and WT littermates. *Lamc1$^{cEQ/+}$* heterozygotes were mated with *Tie2-cre* mice to obtain *Lamc1$^{cEQ/+}$;Tie2-cre* mice. *Lamc1$^{cEQ/+}$;Tie2-cre* male mice were mated with *Lamc1$^{cEQ/+}$* mice to obtain *Lamc1$^{cEQ/cEQ}$;Tie2-cre* and control *Lamc1$^{cEQ/cEQ}$* littermates.

The mice were kept in a specific pathogen-free environment under stable conditions of temperature and light (lights ON at 08:00

and OFF at 20:00). All mouse experiments were performed in compliance with our institutional guidelines and were approved by the Animal Care Committee of Osaka University.

## Ex vivo blastocyst culture

E3.5 blastocysts were flushed from the uterus with flushing-holding medium (Lawitts & Biggers, 1993), washed with potassium simplex optimization medium (Lawitts & Biggers, 1993), and cultured in a hanging drop of potassium simplex optimization medium at 37°C for 48 h under 5% $CO_2$.

## Genotyping

Genomic DNAs were extracted with DirectPCR Lysis Reagent (Viagen Biotech) at 55°C overnight and then heat-denatured at 95°C for 10 min.

The genotypes of $Lamc1^{EQ}$ postnatal mice and whole-mount embryos were determined by genomic PCR using the primer pair 5′-AAGCAGGAGGCAGCCATCATGGACT-3′ and 5′-GGAAGATGCCGTGACT-TCAGGCAAA-3′. WT DNA and $Lamc1^{EQ}$ DNA both gave PCR products of approximately 400 bp. The PCR products were digested with TaqI and separated by 1.8% agarose gel electrophoresis. Although the PCR product derived from the WT allele was digested into bands of 280 and 120 bp, the PCR product derived from the $Lamc1^{EQ}$ allele remained undigested because the γ1 EQ mutation abolished the TaqI site.

Because it is difficult to isolate embryonic tissues without contamination by surrounding maternal tissues, the genotypes of sectioned embryos were determined by their integrin-binding activity in situ (Kiyozumi et al, 2014). For sectioned E5.5–E7.5 embryos, the genotypes were determined by in situ binding of recombinant integrin α7x2β1. Embryos that showed integrin α7x2β1 binding to anti-laminin immunoreactive sites were genotyped as WT or $Lamc1^{EQ/+}$, whereas those that did not show α7x2β1 binding were genotyped as $Lamc1^{EQ/EQ}$. As $Lamc1^{EQ/+}$ mice appeared normal, we did not distinguish $Lamc1^{EQ/+}$ from WT after genotyping by in situ integrin binding and their genotypes are presented as WT or $Lamc1^{EQ/+}$ in the relevant figures.

The genotypes of $Lamc1^{cEQ}$ mice were determined by genomic PCR using the primer pair 5′-TTACCAAGTCACCTTCTTCAGCATAAGCGA-3′ and 5′-GTACATGCGTGTCTGCATGAATGCCATA-3′. The sizes of the PCR products from the WT and $Lamc1^{cEQ}$ alleles were 234 and 423 bp, respectively.

The genotypes of the *Tie2-cre* transgene were determined by genomic PCR using the primer pair 5′-GTTTCACTGGTTATGCGGCGG-3′ and 5′-TTCCAGGGCGCGAGTTGATAG-3′. The size of the PCR product was 450 bp.

## In situ integrin binding

In situ integrin binding was performed as described (Kiyozumi et al, 2012, 2014). Frozen sections of mouse embryos were blocked with blocking buffer (3% BSA, 25 mM Tris–HCl pH 7.4, 100 mM NaCl, and 1 mM $MnCl_2$) for 30 min at room temperature, and then incubated with 3 µg/ml of recombinant integrin α7x2β1 and rat anti-laminin-α1 mAb (5B7-H1) in blocking buffer at 4°C overnight. The sections

were washed three times with wash buffer (25 mM Tris–HCl pH 7.4, 100 mM NaCl, and 1 mM $MnCl_2$) for 10 min at room temperature, and incubated with 0.5 µg/ml of rabbit anti-Velcro pAb in blocking buffer at room temperature for 2 h. After washing with wash buffer, the sections were incubated with Alexa 488-conjugated goat anti-rabbit IgG and Cy3-conjugated anti-rat IgG. The nuclei were stained with Hoechst 33342. After washing with wash buffer, the sections were mounted in PermaFluor (Thermo Scientific Shandon) and visualized with an LSM510 laser confocal microscope (Carl Zeiss).

## Immunofluorescence

Whole-mount immunofluorescence of blastocysts was performed as follows: Blastocysts were fixed in 4% paraformaldehyde in PBS at 4°C for 10 min. The fixed blastocysts were washed with PBS, permeabilized with 0.1% Triton X-100 in PBS at 4°C for 10 min, and incubated with a rabbit anti-laminin pAb and rat anti-perlecan mAb or rat anti-mouse integrin-α6 mAb (GoH3) diluted in 0.1% Tween-20/PBS (TPBS) at 4°C overnight. The embryos were then washed three times with TPBS for 10 min, and incubated with Alexa 546-conjugated goat anti-rabbit IgG and Alexa 488-conjugated goat anti-rat IgG. The nuclei were stained with Hoechst 33342 for 2 h. After three washes with TPBS for 10 min, the blastocysts were placed in a small drop of PBS covered with mineral oil on a coverslip, and visualized under the LSM510 confocal microscope.

Immunofluorescence of sectioned tissues was also performed. Frozen sections of mouse embryos were fixed in 4% paraformaldehyde in PBS at 4°C for 10 min, blocked with 3% BSA/TPBS at 4°C for 30 min, and incubated at 4°C overnight with the following antibodies diluted in 3% BSA/TPBS: mixture of rabbit anti-laminin pAb and rat anti–cytokeratin-8 mAb (TROMA-I); mixture of rat anti–laminin-α1 mAb, mouse anti-Cdx2 mAb, and rabbit anti-Oct4 pAb; or mixture of rabbit anti-laminin pAb and rat anti–laminin-α5 mAb (M5N8-C8). The sections were washed three times with TPBS for 10 min, and incubated at 4°C for 2 h with the following secondary antibodies: Alexa 488-conjugated goat anti-rabbit IgG and Alexa 546-conjugated goat anti-rat IgG; or Alexa 405-conjugated goat anti-rabbit IgG, Alexa 488-conjugated goat anti-rat IgG, and Alexa 546-conjugated goat anti-mouse IgG. The nuclei were stained with Hoechst 33342 if necessary. After three washes with TPBS for 10 min, the sections were mounted in PermaFluor and visualized using the LSM510 laser confocal microscope.

Frozen sections of 8-wk-old mouse retina were air-dried for 30 min at room temperature, fixed with cold acetone at –30°C for 15 min, washed with TBS, blocked with blocking buffer (1% BSA, 25 mM Tris–HCl pH 7.4, 100 mM NaCl, and 1 mM $MnCl_2$) for 30 min at room temperature, and incubated with 3 µg/ml of recombinant integrins and specified antibodies in blocking buffer at 4°C overnight. The sections were washed three times with wash buffer (25 mM Tris–HCl pH 7.4, 100 mM NaCl, and 1 mM $MnCl_2$) for 10 min at room temperature, and incubated with 0.5 µg/ml of rabbit anti-Velcro pAb and anti–laminin-α5 mAb in blocking buffer at room temperature for 2 h. After washing with wash buffer, the sections were incubated with Alexa 488-conjugated goat anti-rabbit IgG and Alexa 546-conjugated anti-rat IgG. The nuclei were stained with Hoechst 33342. After washing with wash buffer, the sections were mounted in PermaFluor and visualized using the LSM510 laser confocal microscope.

The LSM510 laser confocal microscope was equipped with LD-Achroplan (20×, NA 0.4) and Plan-Neofluar (10×, NA 0.3; 40×, NA 0.75) objective lenses at room temperature. The imaging medium was air. The LSM510 PASCAL software (Carl Zeiss) was used for image collection. Each set of stained samples was processed under identical gain and laser power settings. Each set of obtained images was processed under identical brightness and contrast settings, which were adjusted by the LSM image browser (Carl Zeiss) and ImageJ software (Abràmoff et al, 2004) for clear visualization of BMs. At least four pairs of sections for mutant mice and their control littermates were examined, and similar results were obtained.

### Image analysis

The length of the RM was measured on anti-laminin immunofluorescence-stained images of E5.5 egg cylinders using ImageJ software (see also Fig 3P).

### Statistical analysis

A normal distribution of data was confirmed by normal quantile–quantile plot analysis. Effect size, post hoc analyses for actual statistical power ($1-\beta$), and a priori analyses for required sample sizes under given statistical parameters for two-mean analyses were determined by G*Power version 3.1.9.2 (Faul et al, 2009). Statistical significance was determined by a two-tailed Welch's $t$ test using Microsoft Excel for Mac 2011. A value of $P < 0.05$ was considered to indicate a statistically significant difference. $P$-values are only shown in figures where the actual statistical power was ≥0.8.

## Supplementary Information

## Acknowledgements

The authors thank Non-Profit Organization Biotechnology Research and Development for technical assistance with generating the *Lamc1*[EQ] and *Lamc1*[cEQ] knock-in mice, and Dr. Ayako Isotani of the Research Institute for Microbial Diseases, Osaka University, for manipulation and culture of blastocysts. The authors also thank Alison Sherwin, PhD, from the Edanz Group (www.edanzediting.com/ac) for editing a draft of this manuscript. This work was supported by KAKENHI grants 17082005 and 22122006 (to K Sekiguchi).

## Author Contributions

D Kiyozumi: conceptualization, investigation, methodology, and writing—original draft, review, and editing.
Y Taniguchi: investigation and methodology.
I Nakano: methodology.
J Toga: resources.
E Yagi: resources.
H Hasuwa: methodology.
M Ikawa: methodology.
K Sekiguchi: conceptualization, resources, supervision, funding acquisition, and writing—original draft, review, and editing.

## Conflict of Interest Statement

K Sekiguchi is a founder and shareholder of Matrixome Inc. Y Taniguchi is a project leader at Matrixome Inc. All other authors declare that they have no conflict of interest.

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
