## [Reviewer comments · Life Science Alliance]

Life Science Alliance

Laminin γ 1 C-terminal Glu-to-Gln mutation induces early post-implantation lethality

Kiyotoshi Sekiguchi, Daiji Kiyozumi, Yukimasa Taniguchi, Itsuko Nakano, Junko Toga, Emiko Yagi, Hidetoshi Hasuwa, and Masahito Ikawa
DOI: 10.26508/lsa.201800064

Corresponding author(s): Kiyotoshi Sekiguchi, Osaka University

Review Timeline:	Submission Date:	2018-04-04
	Editorial Decision:	2018-05-04
	Revision Received:	2018-08-26
	Editorial Decision:	2018-09-03
	Revision Received:	2018-09-03
	Accepted:	2018-09-04

Scientific Editor: Dr. Andrea Leibfried

Transaction Report:

Thank you for submitting your manuscript entitled "Laminin γ 1 C-terminal Glu-to-Gln mutation induces early post-implantation lethality" to Life Science Alliance. Your manuscript has now been reviewed by three referees whose comments are included below.

As you will see, the referees overall appreciate the quality and significance of your work. However, several questions were raised regarding results from the in vitro laminin binding assay, and further characterization of the basement membrane was also suggested. We therefore think that further experiments are required to make your manuscript a strong candidate for publication. If you think that the manuscript can be modified according to the constructive input provided by the referees, we would be happy to consider a revised manuscript for publication in Life Science Alliance.

In considering a revised manuscript, we suggest focusing on the following items:

1. Two referees raise questions about whether several different integrins (α 6 β 1, α 5 β 1, α 1 β 1, α 2 β 1) can bind to laminin 111. Further experiments and/or added discussion may be needed to respond to these comments.
2. All three referees suggest further experiments (or added discussion) to gain more insight into alterations of the basement membrane. Responding to these comments in a revised manuscript will likely require the addition of new experiments involving improved imaging studies, as well as new imaging studies or the use of in vitro assays.

Thank you for this interesting contribution to Life Science Alliance. We are looking forward to receiving your revised manuscript.

Reviewer #1 (Comments to the Authors (Required)):

The manuscript deals with the contribution of the laminin gamma1 chain to biological functions of laminin isoforms. Laminins are composed of three chains, an alpha, beta and gamma chain and because of their more restricted distribution and association with genetic defects, the alpha chains are considered to carry the main cell binding domains and to convey the biological function of the laminins. The role of laminin beta and gamma chains is less well studied, and elimination of the laminin gamma 1 chain leads to early embryonic lethality. As laminin gamma 1 can combine with several alpha and beta chains, this early embryonic phenotype is thought to be due to lack of heterotrimer formation and hence absence of a laminin network and formation of Reichert's membrane. Based on data from knockout mice and from in vitro adhesion assays, binding of laminins in Reichert's membrane to surrounding trophoblast cells at this early stage is considered to occur via dystroglycan (and not integrins). However, the group of Sekiguchi and others have shown that 8 amino acids at the C-terminus of the gamma 1 chain, together with laminin alpha 1 chain sequences, are required for recognition by several integrins, suggesting a signaling function also for the laminin gamma 1 chain. It is this latter point that is addressed in the current manuscript.

By mutation of this C-terminal laminin gamma 1 chain sequence and generation of both recombinant proteins for in vitro analyses and knock-in transgenic mice, the manuscript addresses the biological function of the laminin gamma 1 chain in early development,

focusing focus on the laminin 111 and 511 isoforms which predominate at these stages.

The data presented suggest that, like the published complete laminin gamma 1 knockout mouse, mice carrying the mutated laminin gamma 1 chain also die at the peri-implantation stage. However, since the mutated laminin gamma 1 chain can be integrated into a laminin heterotrimer and presumably also secreted normally, the indirect message is that the phenotype is due to loss of binding to specific integrins.

In addition, an endothelial cell specific knock-in of the mutated laminin gamma 1 is generated, however, there are open questions concerning the characterization of this strain.

In general, the manuscript provides interesting new information to the laminin community. However, because of the specialized nature of the topic, the manuscript would benefit from some additional background information to highlight the new aspects, as well as addressing of the specific points outlined below.

1. The manuscript provides interesting new data on which integrins actually bind C-terminal domains of laminins 111 and 511. The data shown in Fig 1 clarify several older studies and shed light on which integrins are actually laminin receptors, however, it also raises some questions, the most important of which is why does recombinant integrin $\alpha 6\beta 1$ not recognize laminin 111? Given the ability of antibodies to integrin $\alpha 6$ to extremely efficiently inhibit adhesion of various cells types to purified native laminin 111, one would have expected a direct interaction between the two.
2. Given the controversy about whether integrin $\alpha 5\beta 1$ can also bind laminin 511 - it would have been interesting to include recombinant $\alpha 5\beta 1$ into the study shown in figure 1.
3. Western Blot in Fig 2D demonstrates a large MWt complex visualized with the pan-laminin antibody; however there are several smaller MWt bands as well that could correspond to either laminin complexes or individual chains which, even in small amounts, could affect interactions with surrounding cells and, hence, the interpretation of the results. To definitively address whether both laminin 111 and 511 with a mutated gamma 1 chain can be secreted normally it may be worthwhile to employ the transfected ES cells or even early embryos radiolabelled with ^{35}S -methionine/cysteine for subsequent immunoprecipitation of labeled laminin heterotrimers using a laminin gamma 1 specific antibody versus a laminin alpha 1 and alpha 5 antibody and separation by reducing gels to see all chains (and their relative ratios).
4. In general, the quality of the sections of peri-implantation embryos is not optimal- normal histology would help visualize the defects more clearly.
5. Fig 2E and F - some explanation that the two major laminin isoforms found in Reichert's membrane are laminin 111 and 511 should be provided here to aid in the understanding of the logic for using the recombinant integrin $\alpha 7\beta 1$ to the sections. How was the recombinant protein visualized - which antibodies were used and how can it be excluded that endogenous integrin $\alpha 7\beta 1$ was not detected?
6. Fig 4 Conditional knock-in driven by *tie2cre* - this would result in expression of the mutated protein but it cannot be sure that the original non-mutated protein is no longer present, given the low turnover of laminins in adult tissues (as shown by in situ hybridization studies). What exactly is the message here?

7. On the basis of the data shown in Fig 1, the conditional KO data shown in Figure 4 would suggest that only integrins avb1 or avb3 are relevant integrins for engagement of laminin 511 in blood vessels- is this what the authors suggest? In this context, testing of integrin a5b1 binding to laminin 511 carrying a normal and mutated gamma 1 chain would be interesting and provide important new information.

Reviewer #2 (Comments to the Authors (Required)):

Laminins are key basement membrane proteins that bind and activate several integrins through their LG 1-3 domains and adjacent beta/gamma C-terminal residues contained in the coiled-coil. Previously, the Sekiguchi laboratory established that these interactions require a glutamic acid residue near the C-terminus of the gamma1 subunit common to most laminins. The new manuscript reports the generation of both constitutive and conditional knock-in mice bearing a gamma1 E:Q mutation that ablates integrin binding, and evaluation of the developmental consequences with these mice. First, the authors show that the mutations ablate a7x2b1 integrin binding to intact recombinant Lm111, and a3b1, a6b1, a7x1b1 and a6b4 binding to Lm511, but not avb3 and avb5 binding to Lm511 (known to be mediated by the short arm L4 domain). Second, and importantly, it was found that loss of the laminin-specific integrin binding leads to disordered extraembryonic development despite the persistence of embryonic laminin/basement membrane assembly. Overall the study has been carefully conducted with the results supporting the conclusions. This study is, in the opinion of this reviewer, an important and significant advance in the field.

Specific Comments:

1. Materials and Methods, page 10. The authors state that they used a FLAG-tagged gamma1 chain in preparation of the Lm111 and Lm511. Where is the FLAG tag located? This is not clear from the text or the references provided. A FLAG tag at the C-terminus would be expected to diminish integrin binding.
2. Figure 1. Soluble integrin binding to recombinant Lm111 and Lm511. The authors did not detect a1b1 or a2b1 binding to Lm111. Yet several authors (1,2) previously detected these integrins in cell adhesion assays using recombinant N-terminal fragments of Lm111. Also, the E:Q mutation completely ablated a3b1 binding to whole Lm511 as measured in Fig. 1. Yamada (3) previously reported that there is a second a3b1 binding locus in the a5 LN domain that one would expect is not subject to the E:Q modification. What are possible explanations for the apparent discrepancies? For example, is the LN integrin-binding rendered cryptic by polymerization (binding of the three LN domains together), or is the binding insufficiently strong to be detected by the assay as used (see below), or does laminin adhesion to the plastic block access to the LN domain?
3. Figure 1. a6b1 integrin binding to recombinant Lm111 was not detected. Lm111 has long been considered a ligand for a6b1. In an earlier study of this lab (4), a6b1-Lm111 integrin binding was measured, albeit with a weaker apparent dissociation constant (9.5 nM for Lm111) compared to Lm511 (0.73 nM). This suggests that the single concentration assay as used in the new study can only detect high affinity binding and will therefore fail to detect mid to low affinity interactions. Therefore, it would be valuable to evaluate the binding at different integrin concentrations as was done in the earlier study. It would also help to learn if cell adhesion has been abolished as a result of the EQ mutation. In any case, the issue needs to be discussed at greater length.
4. Figure 3 and associated text. Persistence of laminin in embryonic basement membranes. Laminin immunostaining was detected in E:Q/E:Q embryos as late as E5.5. (a) One conclusion appears to be that the extraembryonic disorder occurs despite persistence of laminin (and hence basement membrane) assembly. Previous studies have reported this in vitro, e.g. ref.(5) showing that basement membranes can form in the absence of integrin

interactions. Unlike previous studies that removed the common beta1 integrin subunit, removal of all integrin interactions with laminins adds a new dimension to the potential role of integrins in basement membrane assembly. The topic should be discussed in the context of the new findings with addition of relevant references.

5. Possible role(s) of integrin in basement membrane (BM) assembly and structure: (a) How late can one detect laminin in a BM pattern in the EQ-null embryos? (b) Are there ultrastructural changes that occur in the EQ/EQ embryonic basement membranes? (c) Are any of the major non-laminin components no longer synthesized/secreted or no longer recruited to the embryonic BMs in EQ nulls?

6. Another question that arises is whether there is any compensatory increase in other laminin receptors, in particular alpha-dystroglycan in Reichert's membrane (perhaps mirroring what was seen in integrin-null embryoid bodies (5)).

References:

1. Colognato-Pyke, H., O'Rear, J. J., Yamada, Y., Carbonetto, S., Cheng, Y. S., and Yurchenco, P. D. (1995) Mapping of network-forming, heparin-binding, and alpha 1 beta 1 integrin-recognition sites within the alpha-chain short arm of laminin- 1. *J. Biol. Chem.* 270, 9398-9406
2. Ettner, N., Gohring, W., Sasaki, T., Mann, K., and Timpl, R. (1998) The N-terminal globular domain of the laminin alpha1 chain binds to alpha1beta1 and alpha2beta1 integrins and to the heparan sulfate- containing domains of perlecan *FEBS Lett* 430, 217-221
3. Nielsen, P. K., and Yamada, Y. (2001) Identification of cell-binding sites on the Laminin alpha 5 N-terminal domain by site-directed mutagenesis. *J Biol Chem* 276, 10906-10912.
4. Nishiuchi, R., Takagi, J., Hayashi, M., Ido, H., Yagi, Y., Sanzen, N., Tsuji, T., Yamada, M., and Sekiguchi, K. (2006) Ligand-binding specificities of laminin-binding integrins: A comprehensive survey of laminin-integrin interactions using recombinant alpha3beta1, alpha6beta1, alpha7beta1 and alpha6beta4 integrins. *Matrix Biol* 25, 189-197
5. Li, S., Harrison, D., Carbonetto, S., Fassler, R., Smyth, N., Edgar, D., and Yurchenco, P. D. (2002) Matrix assembly, regulation, and survival functions of laminin and its receptors in embryonic stem cell differentiation. *J Cell Biol* 157, 1279-1290.

Reviewer #3 (Comments to the Authors (Required)):

Sekiguchi confirm the importance of the C-terminal residue in the gamma 1 chain of laminin in mediating laminin LM111 and LM511 binding to the alpha 3, 6, and 7 integrins. Based on these results, they generate laminin gamma 1 chain EQ knock-in mice and provide evidence that the mutant mice die in utero between E8.5 and E10.5. Defects in the formation of the parietal sac are observed and the authors provide indirect evidence that these may be due to events following integrin binding rather than defects in basement membrane deposition. They go on to generate conditional Knock in mice, which will provide a very useful tool to the field.

I think that the results presented are in general quite solid and the tools generated will be useful. However, I am not convinced that the defects observed are not due to defective incorporation of gamma 1 chain laminins in the BM. The authors need to stain developmentally staged embryos with gamma 1 antibodies as well as antibodies to alpha3, 6, and 7 integrins. In addition, it would be useful to test the incorporation of the mutant laminins in the BM by using in vitro assays, such as those developed by the Yurchenco's laboratory.

Reviewer #1

1. The manuscript provides interesting new data on which integrins actually bind C-terminal domains of laminins 111 and 511. The data shown in Fig 1 clarify several older studies and shed light on which integrins are actually laminin receptors, however, it also raises some questions, the most important of which is why does recombinant integrin $\alpha 6\beta 1$ not recognize laminin 111? Given the ability of antibodies to integrin $\alpha 6$ to extremely efficiently inhibit adhesion of various cells types to purified native laminin 111, one would have expected a direct interaction between the two.

As pointed out by the reviewer, there are numerous studies that have reported inhibition of cell adhesion to laminin-111 by function-blocking antibodies against integrin $\alpha 6$. It should be emphasized that the laminin-111 used in these studies was mouse laminin-111 extracted from mouse Engelbreth-Holm-Swarm (EHS) tumors, while the laminin-111 we used in the experiments shown in Fig. 1 is recombinant human laminin-111.

We therefore examined the binding activity of integrin $\alpha 6\beta 1$ to mouse laminin-111 purified from EHS tumors and our recombinant human laminin-111. The results have been included in new Supplementary Fig. S2B. The results showed that integrin $\alpha 6\beta 1$ was capable of binding to both mouse and human laminin-111 in a dose-dependent manner, but mouse laminin-111 had significantly more affinity than human recombinant laminin-111 for integrin $\alpha 6\beta 1$, explaining why laminin-111 exhibited the very low binding activity to integrin $\alpha 6\beta 1$ in Fig. 1, despite the existing literature demonstrating efficient inhibition of cell adhesion to laminin-111 by anti-integrin $\alpha 6$ antibodies. We have revised the manuscript to include these new results on page 4, lines 14–20 with new Supplementary Fig. S2B.

2. Given the controversy about whether integrin $\alpha 5\beta 1$ can also bind laminin 511 - it would have been interesting to include recombinant $\alpha 5\beta 1$ into the study shown in figure 1.

We performed in vitro binding assays to determine whether integrin $\alpha 5\beta 1$ binds to laminin-111 and laminin-511. The results are shown in new Fig. 1L and new Supplementary Fig. S2A. No significant binding was detected between integrin $\alpha 5\beta 1$ and laminin-111/-511, whereas integrin $\alpha 5\beta 1$ bound strongly to fibronectin. These additional results have been referred to in the revised manuscript, page 4, lines 9 and 26.

3. Western Blot in Fig 2D demonstrates a large MWt complex visualized with the pan-laminin antibody; however there are several smaller MWt bands as well that could correspond to either laminin complexes or individual chains which, even in small amounts, could affect interactions with surrounding cells and, hence, the interpretation of the results. To definitively address whether both laminin 111 and 511 with a mutated gamma 1 chain can be secreted normally it may be worthwhile to employ the transfected ES cells or even early embryos radiolabelled with ^{35}S -methionine/cysteine for subsequent immunoprecipitation of labeled

laminin heterotrimers using a laminin gamma 1 specific antibody versus a laminin alpha 1 and alpha 5 antibody and separation by reducing gels to see all chains (and their relative ratios).

To address whether both laminin-111 and laminin-511 with a mutated $\gamma 1$ chain can be secreted normally, we transfected human 293 cells with these laminins and analyzed the secreted proteins by immunoblotting under non-reducing conditions. The results are shown in new Supplementary Fig. S3. No significant difference was detected in the amounts of secreted heterotrimers between wild-type and EQ-mutated laminin-111/-511, suggesting that laminin-111 and laminin-511 with the mutated $\gamma 1$ chain can be secreted normally. These results have been referred to in the revised manuscript, page 5, lines 13–19.

The smaller molecular weight bands detected in Fig. 2D would be processed and/or degraded laminins. Generally, proteins in tissue extracts include a fraction of partially degraded proteins, the amounts of which vary depending on the tissue type and extraction protocol. Fig. 2D shows that the majority of laminins in the extract had remained intact and suggested that, together with the immunoblot analyses of laminin-111/-511 secreted from 293 cells, laminins with the mutated $\gamma 1$ chain are secreted normally as $\alpha\beta\gamma$ heterotrimers as is the case with laminin-111 secreted from EHS tumors.

4. In general, the quality of the sections of peri-implantation embryos is not optimal- normal histology would help visualize the defects more clearly.

We employed an in situ integrin overlay assay of sections to genotype early post-implantation embryos. Because the assay requires intact integrin-binding activity in the sections, we had to use unfixed frozen sections. For better histology, paraformaldehyde-fixed and paraffin-embedded sections are suitable, but the integrin binding activity of laminins is lost by paraffin embedding. This is why we used unfixed frozen sections for histology.

5. Fig 2E and F - some explanation that the two major laminin isoforms found in Reichert's membrane are laminin 111 and 511 should be provided here to aid in the understanding of the logic for using the recombinant integrin a7x2b1 to the sections. How was the recombinant protein visualized - which antibodies were used and how can it be excluded that endogenous integrin a7x2b1 was not detected?

There are several reports showing that laminin-111 and laminin-511 are expressed in Reichert's membrane (Sasaki et al. Exp. Cell Res. 275:185-199, 2002; Miner et al. Development 131:2247-2256, 2004). We refer to these reports in the revised manuscript (page 5, lines 29–30).

The recombinant integrin a7x2b1 used in our integrin overlay assays contained a “Velcro” tag consisting of ACID and BASE peptides attached to the C-terminal ends of the extracellular domains of integrin a7x2 and b1 subunits, respectively (Nishiuchi et al. Matrix Biol., 25:189-197, 2006). The integrin a7x2b1 bound to tissue sections was detected with a rabbit anti-Velcro antibody as described in the Materials and Methods (page 15, lines 12–22, in the revised manuscript). The anti-Velcro antibody does not cross-react with integrin a7x2b1.

6. Fig 4 Conditional knock-in driven by tie2cre - this would result in expression of the mutated protein but it cannot be sure that the original non-mutated protein is no

longer present, given the low turnover of laminins in adult tissues (as shown by in situ hybridization studies). What exactly is the message here?

Laminin-g1 EQ knock-in mice offer a potentially valuable tool to elucidate the consequences of cell interactions with basement membrane laminins through integrins, because the mutation ablates binding of g1-laminins to a3, a6, and a7 integrins simultaneously. However, early embryonic lethality of the knock-in mice precludes investigations of the consequences of laminin-integrin interactions beyond embryonic development. Thus, another knock-in mouse has been anticipated, in which the laminin-g1 EQ mutation can be introduced at later stages of embryonic development and in adult mice. This is why we employed the conditional knock-in system. Conditional knock-in by Tie2-cre was chosen to only demonstrate that the conditional knock-in system is effective. Detailed analyses of the resulting mice will be dealt with as a separate study. If this manuscript is accepted for publication, we will distribute our conditional laminin-g1 EQ knock-in mice upon request.

7. On the basis of the data shown in Fig 1, the conditional KO data shown in Figure 4 would suggest that only integrins avb1 or avb3 are relevant integrins for engagement of laminin 511 in blood vessels- is this what the authors suggest? In this context, testing of integrin a5b1 binding to laminin 511 carrying a normal and mutated gamma 1 chain would be interesting and provide important new information.

As described above, the reason why we chose Tie2-cre for conditional knock-in was to ensure the feasibility of this knock-in system. We do not intend to investigate the vascular functions of laminins in these mice.

We examined whether integrin a5b1 binds to laminin-511 and its EQ mutant in vitro. The results are shown in new Fig. 1L and Supplementary Fig. S2A. Integrin a5b1 did not bind to laminin-511.

Reviewer #2

1. Materials and Methods, page 10. The authors state that they used a FLAG-tagged gamma1 chain in preparation of the Lm111 and Lm511. Where is the FLAG tag

located? This is not clear from the text or the references provided. A FLAG tag at the C-terminus would be expected to diminish integrin binding.

We apologize for the insufficient description of recombinant laminin preparation. A FLAG tag was inserted just after the N-terminal signal peptide of the laminin g1 chain to facilitate purification of recombinant proteins. This information has been included in the Materials and Methods of the revised manuscript, page 10, lines 27–29.

2. Figure 1. Soluble integrin binding to recombinant Lm111 and Lm511. The authors did not detect a1b1 or a2b1 binding to Lm111. Yet several authors (1,2) previously detected these integrins in cell adhesion assays

using recombinant N-terminal fragments of Lm111. Also, the E:Q mutation completely ablated a3b1 binding to whole Lm511 as measured in Fig. 1. Yamada (3) previously reported that there is a second a3b1 binding locus in the a5 LN domain that one would expect is not subject to the E:Q modification. What are possible explanations for the apparent discrepancies? For example, is the LN integrin-binding rendered cryptic by polymerization (binding of the three LN domains together), or is the binding insufficiently strong to be detected by the assay as used (see below), or does laminin adhesion to the plastic block access to the LN domain?

Many studies reporting the integrin binding specificities of cell-adhesive proteins, including those indicated by this reviewer, relied on cell adhesion inhibition assays by function-blocking integrin antibodies. However, this approach often leads to confusing, possibly false-positive conclusions, as exemplified by the ligand specificity of integrin a3b1, which was once proposed to bind laminin (i.e., laminin-111), collagen, and fibronectin (Wayner EA & Carter WG, J Cell Biol, 105:1873-1884, 1987; Takada Y et al., J Cell Biochem, 37: 385-293, 1988; Elices MJ et al, J Cell Biol 112: 169-181, 1991), but is now considered to be specific for a3 and a5 chain-containing laminin isoforms (Wayner EA et al, J Cell Biol, 121: 1141-1152, 1993; Kikkawa et al, J Biol Chem, 273: 15854-15859, 1998). It has been shown that transfection of the integrin a3 subunit into K562 or RD cells does not correlate with increased binding to any laminin, collagen, or fibronectin (Weitzman et al. J Biol Chem, 268: 8651-8657, 1993), arguing against the recognition of these extracellular matrix proteins by integrin a3b1. Many cell types express relatively large amounts of integrin a3b1 on their cell surface. Therefore, its occupancy by anti-a3 antibodies may indirectly perturb cell-matrix interactions mediated by other integrin types and/or non-integrin receptors. This is why we chose direct integrin binding assays to determine the specificity of integrin-ligand interactions.

Nielsen and Yamada (3) reported that the N-terminal LN domain of the a5 chain is capable of mediating adhesion of HT1080 cells, which is inhibited by a function-blocking antibody against integrin a3 (J Biol Chem, 276: 10906-10912, 2001). They also showed that two synthetic peptides derived from the a5 LN domain, S2 (GQVFHVAYVLIKL) and S6 (RDFTKATNIRLRLR), are capable of inhibiting cell adhesion to the a5 LN domain, but do not inhibit cell adhesion to full length laminin-511/521, suggesting that other region(s), most likely the C-terminal E8 region, play dominant roles in adhesive interactions of cells with laminin-511/521. Consistent with this notion, our direct integrin binding assays showed that integrin a3b1 binding to laminin-511 was abrogated by the EQ mutation within the C-terminal tail of the g1 chain, despite the a5 LN domain remaining unaltered (Fig. 1D). Thus, it appears likely that the interaction of the a5 LN domain with integrin a3b1, if any, is not strong enough to be detected by the integrin binding assay. Similarly, the binding of a1/a5 LN domains to a1b1/a2b1 integrins may not be strong enough to be detected in our integrin binding assay.

3. Figure 1. a6b1 integrin binding to recombinant Lm111 was not detected. Lm111 has long been considered a ligand for a6b1. In an earlier study of this lab (Nishiuchi MB 2006), a6b1-Lm111 integrin binding was measured, albeit with a weaker apparent dissociation constant (9.5 nM for Lm111) compared to Lm511 (0.73 nM). This suggests that the single concentration assay as used in the new study can only detect high affinity binding and will therefore fail to detect mid to low affinity interactions. Therefore, it would be valuable to evaluate the binding at different integrin concentrations as was done in the earlier study. It would also help to learn if cell adhesion has been abolished as a result of the EQ mutation. In any case, the issue needs to be discussed at greater length.

As pointed out by the reviewer, the binding of integrin a6b1 to laminin-111 was not detected in Fig. 1, despite our previous report on their interaction with a weaker dissociation constant compared with laminin-511 (Nishiuchi et al. Matrix Biol 25: 189-197, 2006). It should be emphasized that the laminin-111 used in our previous study was mouse laminin-111 extracted from mouse Engelbreth-Holm-Swarm (EHS) tumors, whereas we used recombinant human laminin-111 in Fig. 1.

We therefore examined the binding activity of integrin a6b1 to mouse laminin-111 purified from EHS tumors and our recombinant human laminin-111 coated on microtiter plates at increasing concentrations. The concentration of integrin a6b1 was fixed at 30 nM, which is close to the saturating concentration. The results have been included in new Supplementary Fig. S2B and showed that integrin a6b1 was capable of binding to both mouse and human laminin-111 in a dose-dependent manner, but mouse laminin-111 had significantly more affinity than human recombinant laminin-111 for integrin a6b1, explaining why laminin-111 exhibited very low activity for binding to integrin a6b1 in Fig. 1. We have revised the manuscript to include these new results on page 4, lines 14–20 with new Supplementary Fig. S2B.

Several reports show abrogation of cell adhesion by the EQ mutation in the following cell types: integrin a6-transfected K562 cells plated on laminin-511 (Ido et al. J Biol Chem 282: 11144-11154, 2007); human ES and

iPS cells plated on laminin-511 E8 fragment (Miyazaki et al. Nat Commun 3: 1236, 2012); human iPS cells plated on laminin-511 E8 fragment (Nakagawa et al. Sci Rep 4: 03594, 2014).

4. Figure 3 and associated text. Persistence of laminin in embryonic basement membranes. Laminin immunostaining was detected in E:Q/E:Q embryos as late as E5.5. (a) One conclusion appears to be that the extraembryonic disorder occurs despite persistence of laminin (and hence basement membrane) assembly. Previous studies have reported this in vitro, e.g. ref.(5) showing that basement membranes can form in the absence of integrin interactions. Unlike previous studies that removed the common beta1 integrin subunit, removal of all integrin interactions with laminins adds a new dimension to the potential role of integrins in basement membrane assembly. The topic should be discussed in the context of the new findings with addition of relevant references.

The results shown in Fig. 3 indicate that laminins assemble into basement membranes in the absence of laminin-integrin interactions, despite the fact that embryoid bodies derived from b1 integrin-null mouse ES cells fail to develop basement membranes (Aumailley et al. J Cell Sci 113: 259-268, 2000; Li et al. J Cell Biol, 157:1279-1290, 2002). However, Li et al (2002; the reference above) demonstrated that laminin-111 is capable of forming basement membranes in b1 integrin-null embryoid bodies when added exogenously. The basement membrane assembly of exogenous laminin-111 can be blocked by addition of the E3 fragment of laminin-111, but not the E8 fragment, indicating that E3-binding non-integrin receptors (e.g., dystroglycan, syndecan, and sulfated glycolipids) play dominant roles in the basement membrane assembly of laminins, which is consistent with our results shown in Fig. 3. These discussions have been included in the revised manuscript, page 6, line 28 to page 7, line 7.

5. Possible role(s) of integrin in basement membrane (BM) assembly and structure: (a) How late can one detect laminin in a BM pattern in the EQ-null embryos? (b) Are there ultrastructural changes that occur in the EQ/EQ embryonic basement membranes? (c) Are any of the major non-laminin components no longer synthesized/secreted or no longer recruited to the embryonic BMs in EQ nulls?

1. (a) Laminin deposition in a basement membrane pattern was observed in EQ/EQ embryos at least by E5.5, as shown in Fig. 3L, N, and O.
2. (b) It would be interesting to determine whether the basement membrane structure was affected in EQ/EQ embryos at the electron microscopic level. However, strong chemical fixation required for electron microscopy makes genotyping of embryos by in situ integrin overlay difficult because of the loss of integrin binding activity of laminins after fixation.
3. (c) We confirmed that perlecan was also expressed and deposited in the basement membrane of ex vivo-cultured EQ/EQ embryos, as shown in Fig. 3J.

6. Another question that arises is whether there is any compensatory increase in other laminin receptors, in particular alpha-dystroglycan in Reichert's membrane (perhaps mirroring what was seen in integrin-null embryoid bodies (5)).

This is an intriguing question. Our preliminary results showed that the immunofluorescence signal of alpha-dystroglycan did not exhibit any significant increase in E4.5 EQ/EQ embryos flushed out from the uterus, although there was a difference in the signal intensity of pericellular alpha-dystroglycan between wild-type

and integrin b1-null mutant embryos, i.e., alpha-dystroglycan signals were increased in b1-null mutant embryos (ref. 5; Li et al. J Cell Biol, 157: 1279-1290, 2002).

Reviewer #3

Sekiguchi confirm the importance of the C-terminal residue in the gamma 1 chain of laminin in mediating laminin LM111 and LM511 binding to the alpha 3, 6, and 7 integrins. Based on these results, they generate laminin gamma 1 chain EQ knock-in mice and provide evidence that the mutant mice die in utero between E8.5 and E10.5. Defects in the formation of the parietal sac are observed and the authors provide indirect evidence that these may be due to events following integrin binding rather than defects in basement membrane deposition. They go on to generate conditional Knock in mice, which will provide a very useful tool to the field. I think that the results presented are in general quite solid and the tools generated will be useful. However, I am not convinced that the defects observed are not due to defective incorporation of gamma 1 chain laminins in the BM.

The authors need to stain developmentally staged embryos with gamma 1 antibodies as well as antibodies to alpha3, 6, and 7 integrins.

In addition, it would be useful to test the incorporation of the mutant laminins in the BM by using in vitro assays, such as those developed by the Yurchenco's laboratory.

In this study, we performed ex vivo culture of wild-type and *Lamc1*^{EQ/EQ} blastocysts flushed out from the uterus to examine the competence of $\gamma 1$ EQ mutant laminins to assemble into a basement membrane (page 6, lines 23-28). Ex vivo culture of blastocysts is considered to be more physiological than in vitro assays using mouse ES cell-derived embryoid bodies or cultured Schwann cells as employed by Yurchenco et al (Li et al, J Cell Biol, 157: 1279-1290, 2002; McLee et al, J Biol Chem, 282: 21437-21447, 2007). Our results showed that laminin deposition in *Lamc1*^{EQ} mutant blastocysts after 48 h of ex vivo cultivation was comparable to that in wild-type blastocysts at the basement membrane of the mural trophoctoderm (Fig. 3I, J) that becomes the parietal yolk sac in later developmental stages. We also observed continuous thick deposition of laminins at the parietal yolk sac basement membrane in E5.5 embryos that had just implanted in the uterine wall (Figure 3O). These findings further supported our conclusion that the defects observed in *Lamc1*^{EQ/EQ} embryos are not caused by defective incorporation of $\gamma 1$ EQ mutant laminins, but due to the defects in laminin-integrin interactions. This conclusion is consistent with a previous report from Yurchenco's laboratory (Li et al. J Cell Biol, 157: 1279-1290, 2002), in which they demonstrated that exogenously added laminin-111 is capable of basement membrane assembly in embryoid bodies derived from $\beta 1$ integrin-null mouse ES cells, indicating that laminins can assemble into basement membranes in the absence of laminin- $\beta 1$ -integrin interactions. Nevertheless, we cannot exclude the possibility that the $\gamma 1$ EQ mutation may affect basement membrane assembly of laminins, which is not discernible at the level of conventional histology, thereby contributing to the phenotype of *Lamc1*^{EQ/EQ} embryos. We therefore referred to this possibility and discussed the competence of $\gamma 1$ EQ mutant laminins for basement membrane assembly in the revised manuscript, page 6, line 28 to page 7, line 7.

We thank the reviewers for their insightful comments that helped us to significantly improve the manuscript with additional data. We hope that the revised manuscript will now be considered acceptable for publication in *Life Science Alliance*.

2nd Editorial Decision

September 3, 2018

Thank you for submitting your revised manuscript entitled "Laminin $\gamma 1$ C-terminal Glu-to-Gln mutation induces early post-implantation lethality". Your study has now been seen by one of the original referees (comments included below) and as you will see this person finds that all concerns have been addressed. We would therefore be happy to publish your paper in *Life Science Alliance* pending final revisions necessary to meet our formatting guidelines.

- > Please make sure that the corresponding author has an ORCID number listed in our system
- > Please include a callout to Fig1 panel M
- > We noticed that a couple of the blots presented are of rather low resolution; in case you have higher resolution versions available we would encourage you to include them.

Thank you for this interesting contribution, we look forward to publishing your paper in *Life Science Alliance*.

Reviewer #2 (Comments to the Authors (Required)):

The manuscript of Kiyozumi and colleagues in the laboratory of Kiyotoshi Sekiguchi makes a substantial advance to our understanding of the role of laminin-integrin binding in development of the early embryo. Furthermore, the generation of a conditional E:Q mutation in the laminin gamma1 chain should allow investigators to probe laminin-integrin contributions in other tissues.

The revisions substantially improve the manuscript, increasing clarity and answering questions raised by the reviewers.